# Evaluation of the Organisation of the COVID-19 Vaccination Process by the Teachers in a Region of Poland

**DOI:** 10.3390/vaccines11101619

**Published:** 2023-10-20

**Authors:** Tadeusz Jędrzejczyk, Anna Tyrańska-Fobke, Daniel Ślęzak, Weronika Kamińska, Mariusz F. Kaszubowski, Agnieszka Bem, Marlena Robakowska

**Affiliations:** 1Department of Public Health & Social Medicine, Faculty of Health Sciences with the Institute of Maritime and Tropical Medicine, Medical University of Gdańsk, 80-210 Gdańsk, Poland; tadeusz.jedrzejczyk@gumed.edu.pl (T.J.); mrobakowska@gumed.edu.pl (M.R.); 2Department of Medical Rescue, Faculty of Health Sciences with the Institute of Maritime and Tropical Medicine, Medical University of Gdańsk, 80-210 Gdańsk, Poland; slezakdaniel@gumed.edu.pl; 3Department of Medical Sociology & Social Pathology, Faculty of Health Sciences with the Institute of Maritime and Tropical Medicine, Medical University of Gdańsk, 80-210 Gdańsk, Poland; kaminskaver@gmail.com; 4Department of Statistics and Econometrics, Faculty of Management and Economics, Gdańsk University of Technology, 80-233 Gdańsk, Poland; markaszu1@pg.edu.pl; 5Department of Corporate and Public Finance, Wroclaw University of Economics and Business, 53-345 Wrocław, Poland; agnieszka.bem@ue.wroc.pl

**Keywords:** COVID-19 vaccine, vaccination strategies, occupational group, Poland

## Abstract

The aim of this study was to assess the organisational process of vaccination within the National Vaccination Programme against COVID-19 (NVP) in the professional group of teachers in Pomeranian Province, Poland. The main goal of the survey was to assess the quality of planning and executing of the NVP and to find a correlation between social and employment placements with the level of perception of chosen quality indicators of the NVP. The presented cross-sectional survey was conducted among 4622 teachers from all levels of education in public and non-public institutions who received the SARS-CoV-2 virus vaccination campaign with the vaccine from AstraZeneca as part of the NVP. The survey was conducted using an original, self-designed questionnaire prepared for this study and distributed to teachers in the form of an online survey via email. Bayesian logistic and linear regression were used to estimate the relationship between predictors and dependent variables. Age was the main factor associated with the performance assessment of the vaccination centre (log[BF] = 0.86–16.88), while gender was the main factor associated with the assessment of NVP (log[BF] = 3.15–10,6). The evaluation of the vaccination registration process (log[BF] = −7.01–50.26) and the evaluation of the information received on the management of post-vaccination reactions (log[BF] = −2.22–65.26) were significant parts of the NVP. It is crucial to tailor information messages to the age and gender of the recipients and to ensure the quality of the information provided by medical personnel, in particular the possible occurrence of vaccination reactions and how to deal with them.

## 1. Introduction

The projected sizes of the populations proposed for COVID-19 vaccination varied by geographical region (demographic structure). In total, 68.4% (95% CI 64.2 to 72.6%) of the global population was willing to be vaccinated against COVID-19, so the population of adults willing to be vaccinated was estimated at 3.7 billion (95% CI 3.2 to 4.1 billion). In contrast, each country had to determine its own preference for vaccine groups and build and evaluate possible vaccination strategies and schedules based on epidemiology and projections of available vaccine doses [1].

The healthcare system requires healthy workers in the system. Thus, the first group to be vaccinated in Poland was medical workers, although the level of acceptance and refusal varied among groups of professionals [2]. However, this was followed closely by a group with a significant impact on the daily lives of Poles, i.e., the broadly defined group of teachers. A common element of policy in most European countries was the prioritisation of healthcare workers and staff and residents in 24-h care centres. Only 10 European countries, including Poland, listed teachers as a priority group for vaccination against COVID-19 [3]. 

### 1.1. Organisation of the Teacher Vaccination Process

Vaccination of those working in educational institutions lasted from 15 January to 15 April 2021. Teachers and university scholars were vaccinated in a comparable way to medical personnel, i.e., they did not self-register via e-registration, but were informed of the details of vaccination service availability by their employers. The group of employees of educational institutions vaccinated in the first instance included: pre-school education teachers including those in the positions of teacher’s aide and tutor’s aide, teachers of classes I–III of primary schools, teachers at special schools and institutions, teachers and instructors of practical vocational training, pedagogical employees of psychological–pedagogical counselling centres, and managers of the institutions. On the other hand, from 15 February 2021 (stage II), teachers at all schools and institutions (all those eligible in each professional group) were registered for vaccination, together with academic teachers [4].

Notification followed a planned process, i.e., teachers notified their employer of their willingness to be vaccinated, the employer filled in a form in the Educational Information System (SIO) or a Government Safety Centre form (in the case of persons caring for children under 3 years of age and nursery workers) and forwarded it to the nodal hospital, an e-referral for vaccination appeared on the Internet Patient Account (IKP), the employer arranged with the nodal hospital for the teachers to be vaccinated, and the employer provided the teachers with the date and place of vaccination. On the other hand, university teachers and staff teaching at universities were notified for vaccination by university rectors through the POL-on system [5]. A diagram of the vaccination process among Polish teachers is presented in Figure 1.

The National COVID-19 Vaccination Programme (NVP) soon became one of the major planning, logistical, and organisational health challenges in Poland. The predictive models prepared covered activities such as the schedule of vaccine delivery, the size of staged recipient groups, or the organisation of vaccination points, which were organised in medical facilities, including general setting, outpatient clinics, and temporary hospitals, as well as additional mobile and temporary points. 

### 1.2. Organisation of the Vaccine Distribution Process

The European Union (EU), under a common purchasing mechanism, contracted for the purchase of vaccines [6]. The volume of orders was proportional to the population. The European Commission negotiated and concluded so-called advance purchase contracts on behalf of the EU Member States [7]. In Poland, five suppliers were allowed, in addition to Sanofi-GSK, and the contract amounted to 2.4 billion PLN. 

In the territory of Poland, the entire logistic process was coordinated by the Material Reserves Agency (ARM), a government institution that performs the tasks that result from the acts on the management of strategic reserves and is subordinate to the Ministry of Climate [8]. ARM cooperated with the Polish Army and the State Fire Service as well as commercial key players—pharmaceutical wholesalers distributing products to pharmacies, hospitals, and clinics. The problem of storage during transport, as well as storage in special conditions, i.e., both the so-called cold supply chain (2–8 °C) and ultra-low temperatures (−75 °C), was solved [9]. Central distribution was also provided with additional equipment required by vaccine manufacturers (syringes, needles, gloves, surgical masks, cotton wools, saline), together with instructions in Polish with information on vaccines and storage conditions and how to carry out vaccinations.

### 1.3. Organisation of Vaccination Centers

The organisation of vaccination centres was linked to the designation of injection sites, these functions were assumed to be performed by a network of medical facilities, including primary and specialist healthcare, vaccination centres, and mobile vaccination teams. Vaccination qualification was a process planned to be performed by a doctor based on an examination and interview with an entry in the medical record, but over time, vaccination qualification could also be performed by other health professions (laboratory diagnosticians, paramedics, physiotherapists, and medical students) after appropriate training. 

The preparation of the vaccination centres themselves for the provision of services was a six-stage process. From the announcement of the invitation to participate in the NVP, with the requirements, through the application of the unit, the processing of the application and the qualification of the subjects in the next stage, the vaccination programme was implemented with sanitary–epidemiological conditions and ensuring its reporting in dedicated information systems for monitoring and analysis of the course, with a final report and termination of the services in temporary arranged facilities.

From the patient’s point of view, the vaccination process itself was based on a referral document, valid for 60 days from the date of issue. These referrals were automatically generated in the P1 system (Electronic Platform for the Collection, Analysis, and Sharing of Digital Resources on Medical Events) in the order of vaccination (specific age groups, professional groups, etc.). Doctors could also issue an individual e-referral for the patient. The appointment process itself was based on a central e-registration system, integrating the individual appointment schedules of the individual vaccination centres. To make an appointment for vaccination, it was possible to use a helpline or electronic registration via the Internet Patient Account (patient.gov.pl), as well as through the facility where the e-referral was issued or directly at the vaccination centre. 

### 1.4. NPV Organisation

Public education and communication were also an important part of the NVP system. The Polish government’s policy in this area focused on building confidence in the vaccination strategy and the vaccine itself, as well as building motivation among the public to decide to proceed with vaccination. Work on these issues involved building a reliable and accessible information platform, a 24-h hotline, the use and publicising of expert knowledge, a broad information campaign, and the distribution of brochures on COVID-19 vaccination.

The monitoring stage of the vaccination process and course was based on the P1 information system (e-referral, e-Vaccination Card), a central e-registration system integrated with individual work schedules of vaccination centres with a link to the supply chain system from the manufacturer to the vaccination centres and the GISCOVID-19 sanitary surveillance system implemented by the Governmental Security Centre (RCB). 

Vaccine safety surveillance was based on the use of current mechanisms and institutions. Surveillance involved a process of control, monitoring, and verification for the investigation of suspected quality defects or adverse vaccination reactions. Supervising institutions include the General Pharmaceutical Inspectorate (GIF), the State Sanitary Inspectorate and the Chief Sanitary Inspectorate (GIS), the Military Sanitary Inspectorate, the National Institute of Public Health—National Institute of Hygiene (NIZP-PZH), or the Office for Registration of Medicinal Products, Medical Devices and Biocidal Products (URPL). Real-time data monitoring and long-term observations made it possible not only to assess the efficacy of vaccines, but above all to really evaluate the effectiveness of vaccination. 

The aim of this study was to conduct an evaluation of the vaccination process in a vocational group of teachers in Pomeranian Province. In our study, we did not focus on the vaccine efficacy and release procedure, as this process is widely described in the literature and in the programme itself. The main point of the survey was assessing the quality of planning and executing of the NVP and to find a correlation between social and employment placements with the level of perception of chosen quality indicators of the NVP. It may be important from the point of view of future planning of vaccine programmes in various professional groups.

## 2. Materials and Methods

The present cross-sectional survey was conducted in April and May 2021 among 4622 teachers from all levels of education (kindergartens, primary schools, secondary schools, universities) in public and non-public institutions in the Pomeranian Voivodeship who benefited from the SARS-CoV-2 virus vaccination campaign with the vaccine from AstraZeneca as part of the NVP. Vaccinations under the NPV were voluntary. According to statistics for Poland, 32,489 teachers were employed in the Pomorskie Voivodeship in 2021 [10]. The surveyed group represents 14.23% of this professional group. The group of teachers was chosen for the study since it represented the second professional group covered by vaccination in Poland after medical personnel. It therefore had a remarkably high priority in the vaccination schedule in Poland, which was not the case until now. As mentioned in the introduction, such a high priority in the vaccination schedule for this professional group was not a common solution used by European countries [3].

The survey was conducted using an original, self-designed questionnaire prepared for this study and distributed to teachers in the form of an online survey via email. The questionnaire created by the authors consisted of 31 questions, mostly closed, single-choice or using Likert scale ratings. The pilot study was conducted on a group of 10 teachers from various levels of education. The results of the pilot study were not included in further analyses. Data were collected using the snowball method by gathering responses from teachers employed in the public and private institutions; the education departments of self-governments of regional, county, and municipal level; then to directors of school supervised by each self-government; and at the end delivered to teachers. All surveyed teachers agreed to participate in the study. Participation in the study was voluntary and did not involve any rewards or compensation. 

Bayesian logistic and linear regression, performed with the statistical package R 4.0.2 and the brms library [11] dedicated to Bayesian regressions, was used to estimate the relationship between predictors and dependent variables. In the first step, a full model containing all selected n predictors was estimated for each dependent variable. Then, n further models were estimated, each with n−1 predictors. To compare the models, the logarithm of Bayes Factor (log[BF]) statistic was used [12]. Values of the log[BF] statistic in favour of the full model log[BF] < 1.1 indicate that the full model is inferior to the model without the given predictor, and thus it should be inferred that the predictor is not related to the dependent variable. Values of the statistic in the range 1.1 < log[BF] < 2.3 indicate a moderate relationship between the predictor and the dependent variable. We considered values in the range 2.3 < log[BF] < 3.4 as suggesting a strong relationship and values of log[BF] > 3.4 as suggesting a strong relationship. Inference of differences between specific levels of the predictor was made based on predicted values calculated from the full model. Approval was obtained from the Independent Bioethics Committee of the Medical University of Gdańsk to conduct the study.

## 3. Results

A total of 4622 teachers took part in the survey. The vast majority of the survey group were women (79.22% vs. 14.04%). The predominant age groups were those aged between 41 and 50 years (34.62%) and residents of large cities (200,000 to 400,000 inhabitants) accounted for 27.65% of the sample. Respondents most often indicated public institutions (93.96%), particularly primary schools (46.02%), as their place of employment. Humanities teachers dominated among the respondents (25.90%). Detailed characteristics of the surveyed population are presented in Table 1.

In Figure 2, the occurrence of associations between the characteristics of the study population and the evaluation of vaccination campaigns within the NVP and the vaccination process at vaccination centres dedicated to teachers in Pomeranian Province is described (Figure 2). 

As indicated by the data presented in the figure above, gender is strongly [logBF 2.3–3.4] associated with the assessment of the NVP recruitment rules and very strongly [logBF > 3.4] associated with the evaluation of the transparency of the NVP rules, the quality of information in the NVP, the openness and transparency of the vaccination process in the NVP, and the way in which the NVP protects vaccinated persons against adverse reactions to vaccination. 

In contrast, age is strongly [logBF 2.3–3.4] associated with the assessment of the level of patient service provided by vaccination centre staff and very strongly [logBF > 3.4] association with the assessment of the history and qualification for vaccination at the vaccination point and the assessment of the information received at the vaccination point about possible post-vaccination reactions. Detailed results of the evaluation of the above elements are presented in Table 2 and Table 3.

The registration process for vaccination at vaccination centres was most often rated very good by the surveyed teachers. Moreover, the interview and qualification for vaccination at the vaccination centres was rated good and very good. The service provided to patients by vaccination centre staff was also mostly rated as very good. The information received at the vaccination centre about possible vaccination reactions and how to deal with them was also rated good and very good by the respondents.

The surveyed teachers were mostly neutral (“hard to say”) in their assessment of all analysed elements of the NVP, such as recruitment rules, transparency of rules, quality of information, openness and transparency of the vaccination process, and the way in which vaccinated persons are protected in case of adverse reactions. 

The relationship between the ratings of the individual elements of the NVP and the assessment of the vaccination process at vaccination centres is presented in Figure 3.

As indicated by the data presented in the figure above, the evaluation of individual NVP components is associated [logBF 1.1–2.3], including strongly [logBF 2.3–3.4] and very strongly [logBF > 3.4], with the evaluation of the registration, interview and vaccination eligibility process, and the evaluation of the information received about adverse vaccine reactions and how to deal with them. The assessment of NVP recruitment rules and assessment of transparency of NVP rules were very strongly associated with the assessment of the vaccination registration process at the vaccination centre [logBF = 50.26]. In contrast, the assessment of the quality of NVP information was strongly associated with the assessment of the interview and vaccination eligibility at the point of vaccination [logBF = 3.22] and very strongly associated with the assessment of the vaccination registration process [logBF = 28.57] and the assessment of the information received at the point of vaccination on how to deal with a vaccination reaction [log = 3.57]. A very strong association was also observed between the assessment of the vaccination registration process at the point of vaccination [logBF = 25.12] and the assessment of the medical history and qualification at the point of vaccination [logBF = 4.37] and the assessment of the openness and transparency of the NVP vaccination process. NVP assessment of how to protect vaccinated persons against adverse vaccine reactions was related to the assessment of information received at the point of vaccination about possible adverse vaccine reactions [logBF = 1.59], but was very strongly related to the assessment of the history and qualification for vaccination at the point of vaccination [lgBF = 9.05] and to the assessment of information received at the point of vaccination about how to deal with adverse vaccine reactions [logBF = 65.26].

The assessment of the quality of NVP vaccine information according to the source of information among the respondents is presented in Table 4.

As shown in Table 4, a correlation between ratings of the quality of information about the vaccine was observed when respondents used information presented on government NVP websites, on public television, on websites, and received from medical personnel (doctor, nurse). Respondents using government NVP websites and public television mostly rated the quality of the information about the vaccine there as 3 and 4 on a five-point scale. Those using websites and information from medical staff mostly rated them at 3 on a five-point scale.

## 4. Discussion

As the results of the study show, age was the main factor associated with the assessment of vaccination centre performance, while gender was the main factor associated with the assessment of NVP. However, when it came to the evaluation of individual elements of the NVP, the evaluation of the vaccination registration process and the evaluation of the information received on the management of post-vaccination reactions were significant factors. 

The lower scores of assessments of NVP given by female teachers can be useful while designing such interventions. Females should thus be considered as a target of focus group analysis of the planned process. The lower assessment of the vaccination service process observed in younger generations of teachers is presumably related to higher expectations of this group. The older part of the population is more experienced with the performance of healthcare services and thus the younger teachers should be considered as a primary target group of the work arrangement planning process at vaccination centres. The other conclusion is that information management at the vaccination point should be organised with regard for the age of NVP beneficiaries. 

Among the gender breakdown of teachers in Poland, it should be noted that almost 80% of teachers are women. Most men are involved in teaching defence preparation, physical education, or mathematics. In contrast, the average age of teachers in Poland is increasing year on year. Currently, it is approximately 47 years old [13]. The surveyed group of teachers presented similar results to the national distribution of age and gender. In the surveyed group, 79.22% are women and the largest age group is 41–50 years old (34.62%). Hence, the results of the research presented may be helpful in explaining the links also found on a wider scale and population.

It should be noted that the management of how vaccines are socially/healthily qualified and distributed to vaccination centres and thus to the final recipient can be compared to the management of the so-called “last mile” in market sectors. This is the final stage of the supply chain, which consists of delivering the order from the warehouse or distribution centre to the final recipient [14], that is, the moment when companies deliver products to so-called regional centres and must deliver them to individual customers, whose organisation is most often deeply differentiated. Efficiency at this stage determines success. 

In the case of COVID vaccination in the USA, the last mile was exceedingly difficult. About a third of ‘customers’ were uncertain whether they wanted the product, and others were concerned that they could not be vaccinated early enough because of limited resources and vaccination planning and management. There was a difficulty in collaboration between public and private healthcare stakeholders [15]. 

Health systems should organise the work of introducing a vaccination programme in four stages. The first task is to gain people’s trust; the second is demand management (communication, prioritisation, and management of the vaccination itself); the third is communication with the public and going beyond just answering ‘Frequently Asked Questions’ and building trust (building an information system on the governments’ website) [16]; the last stage concerns the ‘last mile’ i.e., regional coordination with the relevant institutions. Comparing Israel to the US, for example, Israel having universal insurance and a nationwide digital network integrated into the public health system meant that information about the vaccination process was available to every resident, which arguably influenced the vaccination of about 27% of Israeli citizens, compared to about 4% of the US population at the same time [17].

Although there was regulation in Polish law in the case of pandemic outbreak, there were no official guidelines on how to design, finance, and execute vaccination during the real crises. The NVP was thus a necessity which filled the regulation gap. The quality indicators of both the process design, communication, and performance are some of the factors which can contribute to vaccination coverage success or failure. Teachers, next to healthcare workers, are an important group as the society opinion leaders. The negative assessment of the group is not limited to the standard outcome of deficient performance. The positive experience, on the other hand, can be used to encourage the local and wider school communities to participate in a vaccination programme. Hence, further, in-depth research in this area is very important, especially qualitative research, which will identify the key elements responsible for the success of this type of public health activities. We need to note that the COVID-19 pandemic may have an impact on vaccination against vaccine-preventable diseases (VPDs), and the positive effect associated with this is the high likelihood that the widely recognised need for a coronavirus vaccine may increase people’s overall appreciation of vaccines, which may result in a higher vaccination rate after the pandemic has passed. Governments should take advantage of this effect to act effectively in planning revitalised COVID and post-COVID vaccination programmes [18]. Vaccination in itself saves the lives of 2–3 million people in various age groups annually, and routine vaccination services and outreach programmes have been halted because of the roadblocks imposed due to the escalation of COVID-19 cases worldwide [19].

According to the WHO, lack of trust in vaccination is among the top ten public health risks. However, this may be due to a decline in authority, including medical authority, and a lack of trust in public institutions [19]. Lack of trust in vaccination is not directly attributable to medical reasons, studies indicate that this belief may be an ideology becoming a global problem. These studies show four main reasons for refusal to vaccinate, and these include: lack of trust in public institutions carrying out public health activities (here also influenced by historical circumstances and the experience of Poles), rejection of knowledge in favour of paramedical practices, the spread of fake news, and the collapse of authorities. 

The course of the pandemic changed the attitude of Poles surveyed. A study by the Polish agency CBOS showed that the willingness to vaccinate fluctuated at various times. When asked whether they would like to be vaccinated against COVID-19, the percentage of ‘definitely yes’ and ‘rather yes’ answers was 36% in November 2020, 56% in January 2021, and 34% of respondents in April 2021. The survey also shows that for 61% of respondents, the formulation with which they are vaccinated against COVID-19 is important, which may indicate that we are nevertheless interested in the results of vaccination worldwide (potential complications, withdrawal of certain countries from a particular vaccine) [20].

Among the most important limitations of the presented study, the examined group of teachers may not be fully representative of this professional group in the Pomorskie Voivodship. This is due to limitations in the possibility of conducting the study directly at vaccination points due to constraints related to the epidemiological situation at the time of data collection.

In addition, the evaluation of vaccination centres carried out in the study is unfortunately deeply subjective, as the patient had no comparison with the way the activities were organised at another centre. We can count on the fact that the selected professional group had frequent contact within the group and exchanged experiences. The overall assessment of the whole vaccination process can also be considered, if only due to the characteristics of the teachers themselves (age, range of interests, etc.).

## 5. Conclusions

To build trust in immunization, one of the key features is to target and tailor the message to the audience through appropriate tools. As the results of the presented study indicate, it is important to tailor information messages to the recipients’ age and gender. It is also important to ensure the quality of the information provided by medical personnel, regarding the possible occurrence of vaccination reactions and how to deal with them. 

Neither government nor private organisations can succeed alone. To cross the last mile quickly and fairly, they need to build trust, manage operations well, communicate more effectively, and collaborate with other public and private actors. Vaccination against COVID provides a stress test to help organisations prepare for the other challenges they face.

The successes and failures of pandemic management during COVID-19 should be used for future for planning, organisation, and performance for next probably inevitable pandemic outbreak.. Our findings, along with other quality studies on the vaccination process, have a practical value. The prioritisation of some professional groups as a vaccination subject seems to be reasonable; the main problem in many societies is low acceptance and trust level of immunisation itself. The postulated national quality guidelines for vaccination programmes and process can be useful both in everyday preventive services and during an epidemic crisis. The guidelines should be developed based not only on the point of safety and effectiveness level but also on the perception of the targeted populations.

## Figures and Tables

**Figure 1 vaccines-11-01619-f001:**
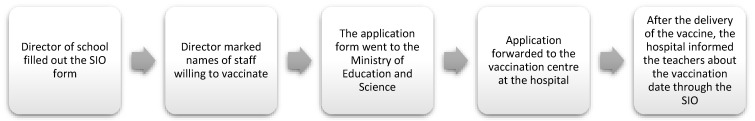
Flow chart of the vaccination process among Polish teachers during the NVP.

**Figure 2 vaccines-11-01619-f002:**
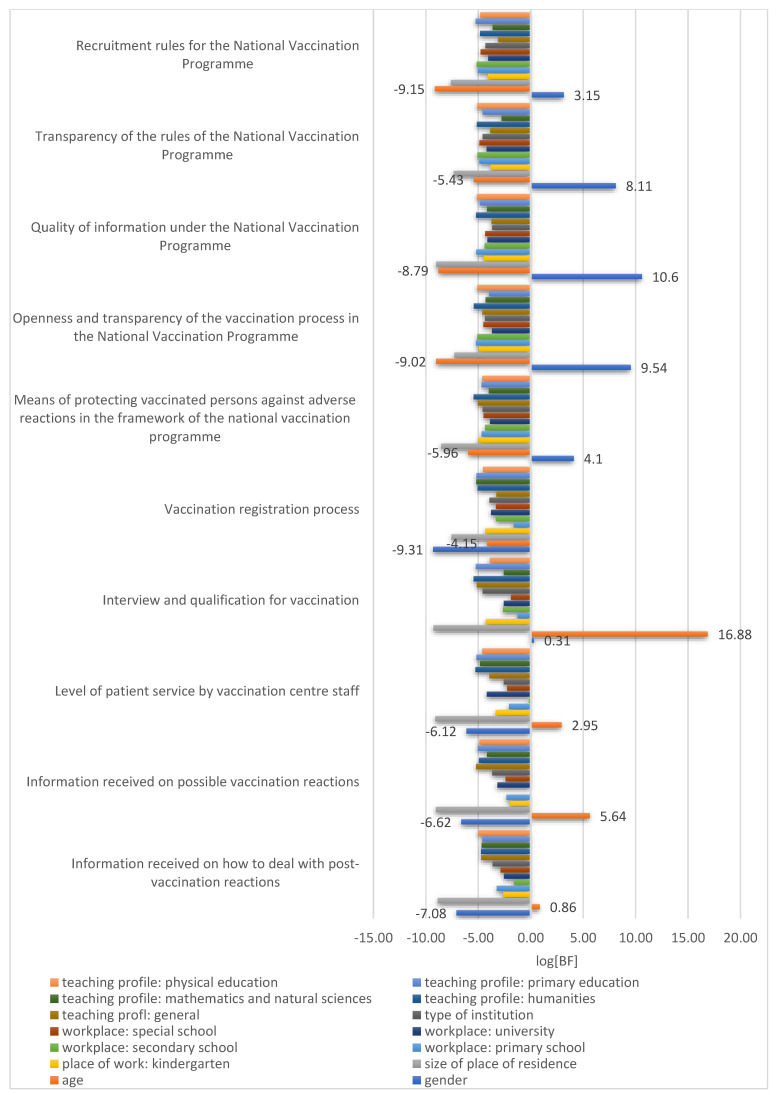
Relationship of selected characteristics of the study population to the evaluation of specific elements of the vaccination process at vaccination centres and the NVP.

**Figure 3 vaccines-11-01619-f003:**
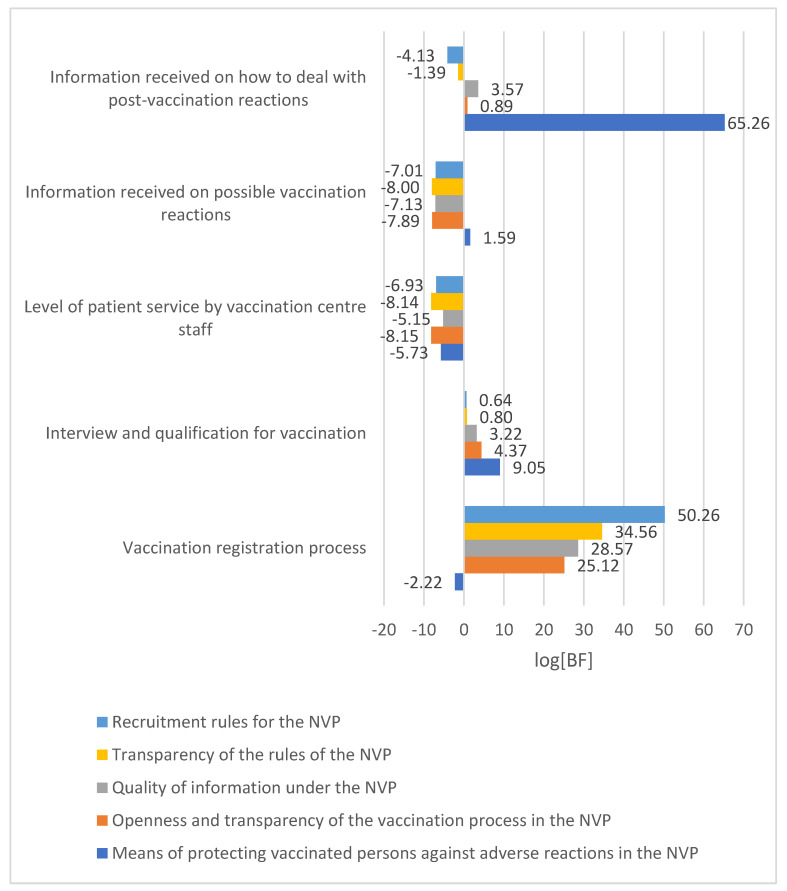
Relationship between the assessment of NVP elements and the vaccination process at vaccination centres.

**Table 1 vaccines-11-01619-t001:** Characteristics of the study population.

Category	Feature	Number	% of 4622
Gender *	Female	3662	79.22%
	Male	649	14.04%
	I don’t want to answer that question	136	2.94%
	Total	4447	96.2%
Age (in years) **	20–30	281	6.08%
	31–40	891	19.28%
	41–50	1600	34.62%
	51–60	1385	29.97%
	61–70	278	6.01%
	Total	4435	95.96%
Size of place of residence *	up to 5000 inhabitants	504	10.9%
	between 5000 and 20,000 inhabitants	189	4.09%
	more than 20,000 to 100,000 inhabitants	1242	26.87%
	over 100,000 to 200,000 inhabitants	779	16.85%
	over 200,000 to 400,000 inhabitants	1278	27.65%
	more than 400,000 inhabitants	455	9.84%
	Total	4447	96.2%
Workplace ***	Kindergarten	404	8.74%
	Primary school	2127	46.02%
	Secondary school	1679	36.32%
	University	58	1.55%
	School for special needs	288	6.23%
	Total	4556	98.86%
Type of institution *	Private	104	2.25%
	Public	4343	93.96%
	Total	4447	96.21
Teaching profile ****	General	566	12.25%
	Humanities	1197	25.90%
	Mathematics and natural sciences	777	16.81%
	Early childhood education	658	14.24%
	Physical education	373	8.07%
	Total	3571	77.27%

Missing data: * 175; ** 187, *** 66, **** 1051.

**Table 2 vaccines-11-01619-t002:** Evaluation of individual elements of the vaccination process at vaccination centres by respondents by age group.

Age (in Years)	Very Bad		Bad		Hard to Say		Good		Very Good		log [BF]
	%	95% BCI	%	95% BCI	%	95% BCI	%	95% BCI	%	95% BCI	
Registration process for vaccination at the vaccination centre											
20–30	2.4	1.36–4.14	6.47	3.75–10.13	9.86	6.26–13.93	36.14	29.21–39.83	45.18	32.7–59.19	−4.15
31–40	2.02	1.18–3.41	5.44	3.29–8.49	8.53	5.58–12.18	34.19	27.12–39.83	49.83	37.81–62.43	
41–50	1.78	1.04–2.99	4.89	2.95–7.58	7.76	5.00–11.15	32.65	25.45–38.01	52.91	41.00–65.3	
51–60	1.67	0.97–2.81	4.57	2.76–7.22	7.38	4.67–10.66	31.78	24.56–37.58	54.55	42.11–66.88	
61–70	1.65	0.93–2.87	4.53	2.66–7.27	7.31	4.48–10.83	31.64	23.94–37.66	54.86	41.79–67.94	
Interview and qualification for vaccination at the vaccination centre											
20–30	3.28	1.95–5.51	8.03	5.01–12.45	14.73	10.2–19.89	39.17	35.48–41.1	34.44	23.71–47.03	16.88
31–40	2.66	1.57–4.33	6.64	4.17–10.23	12.79	8.76–17.56	38.3	33.2–40.85	39.45	28.21–51.93	
41–50	2.13	1.25–3.52	5.43	3.40–8.48	10.94	7.26–15.42	36.52	30.22–40.19	4485	33.17–57.71	
51–60	1.71	1.00–2.84	4.41	2.7–7.00	9.18	5.98–13.42	34.06	26.9–38.99	50.64	38.41–63.27	
61–70	1.37	0.76–2.38	3.56	2.10–5.87	76	4.78–11.63	31.15	23.43–37.56	56.34	42.86–69.00	
Level of patient service by vaccination centre staff											
20–30	2.99	1.67–5.06	5.62	3.44–8.90	7.71	4.84–11.28	39.61	32.44–44.02	43.85	31.77–57.24	2.95
31–40	2.3	1.3–3.86	4.39	2.67–6.81	6.26	3.88–9.14	36.35	28.45–41.88	50.68	38.91–63.34	
41–50	1.9	1.06–3.18	3.68	2.16–5.8	5.32	3.24–7.93	33.49	24.94–39.96	55.58	43.53–68.34	
51–60	1.69	0.96–2.85	3.28	1.92–5.24	4.82	2.92–7.34	31.69	23.21–38.72	58.36	46.25–70.97	
61–70	1.63	0.89–2.84	3.17	1.8–5.11	4.66	2.76–7.28	31	22.26–38.72	59.49	46.19–72.24	
Information received at the vaccination centre on possible vaccination reactions											
20–30	8.91	5.58–14.13	13.64	9.45–18.96	20.49	16.45–23.50	29.85	15.78–31.88	26.95	17.9–37.55	5.64
31–40	7.1	4.53–11.17	11.42	7.85–16.15	18.55	14.42–22.28	30.5	28.19–32.06	32.14	22.32–42.87	
41–50	5.91	3.70–9.49	9.89	6.63–14.41	16.95	12.79–21.11	30.51	28.39–32.06	36.3	25.78–47.88	
51–60	5.21	3.23–8.34	8.9	5.86–13.19	15.75	11.64–20.17	30.29	27.60–32.03	3959	28.21–51.18	
61–70	4.85	2.97–7.95	8.3	5.33–12.67	15.03	10.73–19.88	30.07	26.63–31.90	41.55	29.45–54.27	
Information received on how to deal with post-vaccination reactions											
20–30	10.9	6.99–16.54	14.18	10.09–18.56	22.91	18.96–24.55	27.4	23.37–29.82	24.45	16.73–34.03	0.86
31–40	8.97	5.88–13.57	12.28	8.57–16.36	21.47	15.62–23.78	28.45	25.52–30.41	28.55	20.15–38.83	
41–50	7.79	4.97–11.96	11	7.49–14.97	20.2	14.61–23.08	28.79	26.57–30.41	31.95	22.68–43.46	
51–60	7.07	4.45–10.96	10.16	6.71–14.08	19.2	13.77–22.87	28.9	26.88–30.51	34.27	24.38–46.28	
61–70	6.69	4.06–10.47	9.66	6.29–13.80	18.73	23.37–29.82	28.9	26.73–30.47	35.6	25.27–48.53	

**Table 3 vaccines-11-01619-t003:** Evaluation of individual elements of the NVP by respondents by gender.

Gender	Very Bad		Bad		Hard to Say		Good		Very Good		log [BF]
	%	95% BCI	%	95% BCI	%	95% BCI	%	95% BCI	%	95% BCI	
Recruitment rules for the NVP											
Female	8.03	4.98–12.29	30.23	22.39–37.99	38.3	34.01–40.46	18.99	12.84–26.30	4.34	2.66–6.92	3.15
Male	6.87	4.28–10.85	27.64	19.70–35.84	38.86	35.47–40.73	21.25	14.41–29.14	5.05	3.10–8.09	
I don’t want to answer that question	16.56	9.97–26.12	41.98	34.04–46.85	29.57	21.41–36.80	9.73	5.78–15.69	1.92	1.06–3.42	
Transparency of the rules of the NVP											
Female	8.76	5.47–13.42	29.44	21.98–37.18	37.25	33.09–39.44	19.85	13.62–27.11	4.53	2.79–7.18	8.11
Male	8.67	5.44–13.58	29.24	21.80–36.80	37.31	33.12–39.41	19.96	13.48–27.49	4.61	2.81–7.21	
I don’t want to answer that question	21.78	13.59–33.52	42.1	36.28–45.36	25.76	17.98–33.37	8.36	4.80–13.68	1.61	0.88–2.84	
Quality of information under the NVP											
Female	9.29	6.00–14.07	31.67	24.05–38.83	39.17	33.94–42.19	16.68	11.44–23.41	3.12	1.98–4.94	10.6
Male	7.73	4.92–12.22	28.5	21.15–36.43	40.43	35.91–42.71	19.38	13.22–26.64	3.8	2.24–6.08	
I don’t want to answer that question	22.5	13.60–34.67	43.26	37.90–46.62	25.79	17.27–34.24	6.92	4.03–11.56	1.13	0.62–2.04	
Openness and transparency of the vaccination process in the NVP											
Female	9.12	5.84–14.16	26	19.06–32.88	39.84	36.05–41.90	19.85	13.76–27.11	4.86	2.99–7.71	9.54
Male	8.03	5.13–12.75	24.05	17.30–31.53	40.12	37.15–42.03	21.72	15.22–29.15	5.54	3.42–8.74	
I don’t want to answer that question	22.53	14.24–34.11	38.3	32.96–41.45	28.7	20.32–36.15	8.47	5.03–13.38	1.73	0.98–3.05	
Means of protecting vaccinated persons against adverse reactions in the NVP											
Female	15.38	5.84–14.16	26	19.06–32.88	39.84	36.05–41.90	19.85	13.76–27.11	4.86	2.99–7.71	4.10
Male	8.03	5.13–12.75	24.05	17.30–31.53	40.12	37.15–42.03	21.72	15.22–29.15	5.54	3.42–8.74	
I don’t want to answer that question	22.53	14.24–34.11	38.3	32.96–41.45	28.7	20.32–36.15	8.47	5.03–13.38	1.73	0.98–3.05	

**Table 4 vaccines-11-01619-t004:** Assessment of the quality of NVP vaccine information by source of information among respondents.

Vaccine Information Sources	1		2		3		4		5		log [BF]
	%	95% BCI	%	95% BCI	%	95% BCI	%	95% BCI	%	95% BCI	
From government websites dedicated to the NVP	1.72	0.99–2.91	8.50	5.35–13.29	37.79	29.45–44.88	37.84	30.25–43.03	13.99	8.91–21.57	30.52
From public TV	1.88	1.07–3.14	9.07	5.76–14.17	38.95	30.65–45.80	36.85	28.66–42.59	13.15	8.11–20.41	5.26
From commercial TV	2.27	1.33–3.84	10.78	6.82–16.39	41.75	34.00–47.26	34.06	25.82–40.73	11.08	6.91–17.33	−5.47
From online media	2.69	1.58–4.47	12.45	8.08–18.83	43.97	37.06–48.20	31.37	23.38–38.60	9.43	5.89–14.82	5.66
From the press	2.30	1.34–3.79	10.85	6.83–16.50	41.94	34.00–47.33	33.87	25.78–40.45	10.90	6.83–17.06	−5.37
From a doctor, nurse	2.65	1.58–4.48	12.38	7.92–18.50	43.90	36.87–48.14	31.49	23.55–38.80	9.51	5.92–15.07	4.43
From friends, family or colleagues	2.59	1.51–4.25	12.06	7.7–17.85	43.48	36.20–48.01	32.03	23.90–39.10	9.75	6.13–15.44	0.25

## Data Availability

The data are unavailable due to privacy or ethical restrictions.

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
