# Peer review of "Evaluation of the Organisation of the COVID-19 Vaccination Process by the Teachers in a Region of Poland"

_vaccines, 2023, doi:10.3390/vaccines11101619_

Round 1
Reviewer 1 Report
Review
Evaluation of the organisation of the COVID-19 vaccination process by the teachers in a region of Poland.
I think this paper has something to offer, but I can't quite figure out what the authors and the data are trying to say. Much of this could be due to the authors writing in a second language, but it’s not enough where I can tell one way or the other.
My two main problems go to the heart of the issue:
-
What is the point of the paper? I think it’s using a survey to figure out how to best run a vaccination program based on the NVP.
-
The major conclusion seems to be in lines 24-26 of the abstract, but I’m not sure: a) what these two conclusions mean—the conclusions about age and gender. And b) so what? If this information WAS clear, what does it suggest? Mean? What should we do with it?
These suggestions are address problematic areas and may help, additionally, clarify the major points 1 and 2 above:
-
In the abstract and text the authors say they use an original survey from the internet. If it’s original, it’s not from the internet!
-
The end of the sentence 'although perhaps not always prepared for this’ (lines 44-45) is unclear.
-
Line 49: not clear what this means: that the employees were notified by their employers
-
Line 50: do you mean educational institutions?
-
Several acronyms, such as POL and POZ, are used but not defined.
-
Line 102: what does ‘liquidation of the point’ mean?
-
It would be very helpful to have a flow diagram of the vaccination process that is attempted to be described in the introduction.
-
Line 149: employed in the following institutions. But then no institutions follow.
-
The figures, all of them but especially Tables 2 and 3 are a mass of numbers that are very difficult to understand. Consider optional ways of displaying the data where it’s more clear what the point is and why.
-
Lines 279-286: are you talking about the Polish program now? I don’t understand this paragraph at all.
-
Is the age and gender distribution of your participants a reflection of those distributions of K-12 teachers overall? And does this help explain the associations you find, rather than saying anything about Covid and vaccinations?
-
Again, in the conclusions, Lines 330-337, who cares? What does it matter? What are your recommendations based on your findings (once you make it clearer what those actually are)?
Needs much work as noted.
Author Response
Thank you for your valuable comments. Please see the attachment.
Response to Reviewer 1 Comments
1. Summary
Thank you very much for taking the time to review this manuscript. Please find the detailed responses below and the corresponding revisions in track changes in the re-submitted files.
3. Point-by-point response to Comments and Suggestions for Authors
Comments 1: I think this paper has something to offer, but I can't quite figure out what the authors and the data are trying to say. Much of this could be due to the authors writing in a second language, but it’s not enough where I can tell one way or the other. My two main problems go to the heart of the issue:
Response 1: Thank you for your valuable comments. We have made extensive corrections to the content of the entire manuscript for better clarity. We have applied substantive as well as linguistic corrections.
Comments 2: 1.What is the point of the paper? I think it’s using a survey to figure out how to best run a vaccination program based on the NVP.
Response 2: Thank you for your valuable comments. The point of the survey was assessing quality of planning and executing of NVP and to find correlation between social and employment placements with the level of perception of chosen quality indicators of the Program. We have added this to abstract and to the introduction to emphasize this point.
Comments 3: The major conclusion seems to be in lines 24-26 of the abstract, but I’m not sure:
a) what these two conclusions mean—the conclusions about age and gender. And b) so what? If this information WAS clear, what does it suggest? Mean? What should we do with it?
Response 3: Thank you for your valuable comments. The lower scores of assessment of NVP given by female teachers can be useful while designing such interventions. The females should be thus be consider as a target of focus group analysis of planned process. The lower assessment of the vaccination service process observed in younger generations of teachers are probably related to higher expectations of this group. The older part of the population is more
experienced with way of performance of healthcare services and thus should the younger teachers are to be considered as a primary target group of work arrangement planning process at vaccination centers. The other conclusion is information management at vaccination point which should be targeted with regards for the age of NVP benefiters. We have added this to discussion section to emphasize this point.
Comments 4: In the abstract and text the authors say they use an original survey from the internet. If it’s original, it’s not from the internet!
Response 4: Thank you for your valuable comments. This was our linguistic error and lack of precision in the description. As authors, we prepared an original questionnaire for the purpose of the study, which was posted on the web portal and distributed among teachers by e-mail. We have clarified this both in abstract and in the text of our manuscript.
Comments 5: The end of the sentence 'although perhaps not always prepared for this’ (lines 44-45) is unclear.
Response 5: Thank you for your valuable comments. We have changed this sentence to the following: “Thus, the first group to be vaccinated in Poland was medical workers, although the level of acceptance and refusal was varied among groups of professionals [2].”
Comments 6: Line 49: not clear what this means: that the employees were notified by their employers
Response 6: Thank you for your valuable comments. We have changed this sentence to the following: Teachers and university scholars were vaccinated in a similar way to medical personnel, i.e. they did not self-register via e-registration, but were informed of the details of vaccination services availability by their employers.
Comments 7: Line 50: do you mean educational institutions?
Response 7: Thank you for your valuable comments. Yes, it should be as you indicated. This was our linguistic error. We have changed this form to the correct form.
Comments 8: Several acronyms, such as POL and POZ, are used but not defined.
Response 8: Thank you for your valuable comments. The name POL-on is not an acronym, but the proper name of the government's tele-informatics system for collecting data on science and higher education by enabling universities to report remotely. On the other hand, the use of the abbreviation POZ is actually our oversight. We have changed the sentence to the following: “The predictive models prepared covered activities such as the schedule of vaccine delivery, the size of staged recipient groups or the organization of vaccination points, which were organized in medical facilities, including general setting, outpatient clinics and temporary hospitals, as well as additional mobile and temporary points.”
Comments 9: Line 102: what does ‘liquidation of the point’ mean?
Response 9: Thank you for your valuable comments. We have changed the sentence to the following to be more specify: “In the next stage, the vaccination programme was implemented with sanitary-epidemiological conditions and ensuring its reporting in dedicated information systems for monitoring and analysis of the course, with a final report and termination of the services in temporary arranged facilities.”
Comments 10: It would be very helpful to have a flow diagram of the vaccination process that is attempted to be described in the introduction.
Response 10: Thank you for your valuable comments. We have added the flow diagram in the introduction.
Comments 11: Line 149: employed in the following institutions. But then no institutions follow.
Response 11: Thank you for your valuable comments. We have clarified the indicated sentence and added sentence as following: “The education departments of self-governments of regional, county and municipal level, then to CEOs of school supervised by each of self government and at the end delivered to teachers.”
Comments 12: The figures, all of them but especially Tables 2 and 3 are a mass of numbers that are very difficult to understand. Consider optional ways of displaying the data where it’s more clear what the point is and why.
Response 12: Thank you for your valuable comments. Another reviewer also pointed out the legibility of the tables and recommended reformatting them, which we did according to his suggestion.
Comments 13: Lines 279-286: are you talking about the Polish program now? I don’t understand this paragraph at all.
Response 13: Thank you for your valuable comments. We have changed the whole paragraph as following: “Although there were regulation in Polish low in the case of pandemic outbreak there was no official guidelines how to design, finance and execute vaccination during the real crises. NVP was than a necessity which fill up the regulation gap. The quality indicators of both the process design, communication and performance is one of the factor which can
contribute to vaccination coverage success or failure. The teachers, next to healthcare workers are important group as the society opinion leaders. The negative assessment of the group is not limited to standard outcome of poor performance. The positive experience, for the other hand, can be used to encourage the local and wider school communities to participate in
vaccination program.”
Comments 14: Is the age and gender distribution of your participants a reflection of those distributions of K-12 teachers overall? And does this help explain the associations you find, rather than saying anything about Covid and vaccinations?
Response 14: Thank you for your valuable comments. We have clarified and added sentence as following: “Among the gender breakdown of teachers in Poland, it should be noted that almost 80% of teachers are women. Most men are involved in teaching defense preparation, physical education or mathematics. By contrast, the average age of teachers in Poland is increasing year on year. Currently, it is approximately 47 years old [12]. Similar to the national distribution by age and gender is presented by the surveyed group of teachers. In the surveyed group, 79.22% are women and the largest age group is 41-50 years old (34.62%). Hence, the results of the presented research may be helpful in explaining the links found also on a wider scale and population.”
Comments 15: Again, in the conclusions, Lines 330-337, who cares? What does it matter? What are your recommendations based on your findings (once you make it clearer what those actually are)?
Response 15: Thank you for your valuable comments. The successes and failures of pandemic management during COVID-19 should be used to next and probable uninheritable next pandemic outbreak planning, organization and performance. Our findings, along with other quality studies on vaccination process have a practical value. The prioritization of some professional group as a vaccination subject seems to be reasonable, the main problem in many societies is low acceptance and trust level for immunization itself. The postulated national
quality guidelines for vaccination programs and process can be useful both in everyday preventive services and while epidemic crisis. The guideline should be developed based not only from the point of safety and effectiveness level but also by perception of the targeted populations. We have added this to conclusion to emphasize this point.
4. Response to Comments on the Quality of English Language
Point 1: Needs much work as noted.
Response 1: We subjected the content of our manuscript to intensive linguistic editing.
5. Additional clarifications

Reviewer 2 Report
I read with a lot of interest this manuscript by Jędrzejczyk et.al. who evaluated the organisation of the COVID-19 vaccination process among 4622 teachers. This is a cross-sectional study conducted via an online survey. The topic is extremely useful for stakeholders who draw up vaccination policy in modern society and also for researchers who wish to deepen their knowledge in this field. The study is well written in a large cohort. The statistical analysis is very good and the conclusions are supported by the results.
However, the paper suffers from a serious methodological weakness and if it cannot be fixed I cannot but reject the paper. The Methods section is very short and needs updating. The authors said that they used “questionnaire available on the Internet”. This is very problematic. Also, I assume there is no peer reviewed articles that used this questionnaire, otherwise the author would have cited the references. I think the author needs to show that they have done some reliability studies for the questionnaire. Also, they should have done a pilot study. Did they translated the questionnaire to the local language? If so, how they confirmed that they did a good translation. The authors could also do some crude psychometrics like using Cronbach’s alpha to confirm that the revised, abbreviated measure items within each domain do show some coherence.
Author Response
Thank you for your valuable comments. Please see the attachment.
Response to Reviewer 2 Comments
1. Summary
Thank you very much for taking the time to review this manuscript. Please find the detailed responses below and the corresponding revisions in track changes in the re-submitted files.
3. Point-by-point response to Comments and Suggestions for Authors
Comments 1: I read with a lot of interest this manuscript by Jędrzejczyk et.al. who evaluated the organisation of the COVID-19 vaccination process among 4622 teachers. This is a crosssectional study conducted via an online survey. The topic is extremely useful for stakeholders who draw up vaccination policy in modern society and also for researchers who wish to deepen their knowledge in this field. The study is well written in a large cohort. The statistical analysis is very good and the conclusions are supported by the results.
Response 1: Thank you for your valuable comments.
Comments 2: However, the paper suffers from a serious methodological weakness and if it cannot be fixed I cannot but reject the paper. The Methods section is very short and needs updating. The authors said that they used “questionnaire available on the Internet”. This is very problematic. Also, I assume there is no peer reviewed articles that used this questionnaire, otherwise the author would have cited the references.
Response 2: Thank you for your valuable comments. This was our linguistic error and lack of precision in the description. As authors, we prepared an original questionnaire for the purpose of the study, which was posted on the web portal and distributed among teachers by e-mail. As the study used an original questionnaire, there are no published peer-reviewed articles to cite. We have clarified this both in abstract and in the text of our manuscript.
Comments 3: I think the author needs to show that they have done some reliability studies for the questionnaire. Also, they should have done a pilot study.
Response 3: Thank you for your valuable comments. The reliability of the questionnaire has not been tested. However, a pilot study was conducted on a group of 10 teachers from different levels of education. The results of the pilot study came out very well. The questionnaire did not need to be revised or amended. The results of the pilot study were not included in further analyses. We have added this in the text of our manuscript.
Comments 4: Did they translated the questionnaire to the local language? If so, how they confirmed that they did a good translation.
Response 4: Thank you for your valuable comments. The questionnaire was prepared from the outset in Polish and did not require translation into the local language.
Comments 5: The authors could also do some crude psychometrics like using Cronbach’s alpha to confirm that the revised, abbreviated measure items within each domain do show some coherence.
Response 5: Thank you for your valuable comments. The tool used was primarily designed to measure respondents' opinions and assessments of the various elements of NPV. Hence, psychometric tests such as Cronbach's alpha coefficient, among others, were not conducted.
4. Response to Comments on the Quality of English Language
Point 1: English language fine. No issues detected.
Response 1: Thank you for your valuable comments.
5. Additional clarifications

Reviewer 3 Report
You can find attached the review file
The manuscript entitled "Evaluation of the organisation of the COVID-19 vaccination process by the teachers in a region of Poland " investigates the opinions on the vaccination campaign of teachers in Poland. The population that was analysed is a very specific subgroup, yet the sample is relatively large. The topic is also still very timely. The results obtained are interesting and potentially applicable in other contexts, such as flu vaccination.
There are some issues that should be described in the methods section:
-Why did the researchers choose the teachers?
-teachers were found through the snowball effect, however at the beginning how did the researchers contact the first teachers?
-how many teachers did the researchers contact in total? So, what percentage of the teachers contacted answered the questions in the survey?
-did all the teachers give their consent?
-was participation compulsory? Was there a reward for participation?
-did the researchers follow any template or guidelines for writing this quantitative article?
-is there a particular reason why the researchers chose that region of Poland to conduct the research? Where is the region located and what is peculiar about it compared to others?
-Why did the researchers choose that questionnaire?
In the discussion section there is no interpretation of the meaning of the results, and consequently the comparison of your own results with those in literature.
The results are presented in the Results section, while in the discussion a commentary of the salient results should be provided without rewriting them again, both positive and negative, to try to identify possible meanings, patterns and relationships between your results and, if possible, between the results of other studies similar to yours.
For example:
-how come in Figure 1 age and gender turned out to be so relevant?
-in table 2 is there a directly proportional relationship between age of teachers and percentage of "very good" response for some items? How so?
-in table 3 many teachers answered: "hard to say." how come? Did they ever pay attention to the quality of information or transparency of rules?
-what do the results obtained in figure 2 mean?
-do you think in table 4 some sources of information received quality unexpectedly? Such as online media having the same score as doctor, nurse? Or TV seems to give same opinions as government websites?
Are there any other questionnaires in the literature on this subject about teachers? Knowing how to argue one's own results and those of other authors is important to put one's work in the context of the scientific literature (and thus the validity of one's work), whether the results agree or disagree.
What are the implications of the study? In discussion the authors can reason about potential practical applications of the results.
In the methods the authors vaguely mention where the questionnaire comes from. Can you include the website from which you obtained the questionnaire? Who designed it? In what areas was it tested? Why did the authors consider that questionnaire appropriate for your research question?
In the introduction the authors describe the NVP; was vaccination mandatory or voluntary? Could the authors specify that in the methods section?
In the introduction the authors need to stimulate the reader's interest and to highlight the importance of their work. Explain why research such as yours would be important in the past, current, or future context.
From your evaluation of your results and others in the literature, have you identified gaps in knowledge and that could be explored? The authors might, for example, suggest in discussion that future research should investigate teachers' opinions through qualitative research methodology.
The results show that gender is associated with NVP evaluation. In what way? For example: in Figure 1, what difference would there be between males and females in the assessment? In table 3 are there any differences between genders?
Normally in methods you do not describe how statistical tests work; in lines 154-161, the authors seem to be describing statistical concepts. I suggest the authors put the concepts expressed on log[BF] for example in this other way: "we considered values of log[BF] > 3.4 as suggesting a very strong relationship..".
Among the limitations of the study, the authors report that the questionnaire is not representative of the group of teachers in the region due to the impossibility of conducting the questionnaire in situ during vaccinations. As a first piece of advice, do not say "is not a fully representative group", but instead it is better to say, "may not be fully representative". In this way you are reporting the limitation of your work without making it appear that you are demolishing the validity of your own sample.
Since the authors did not report in the methods, however, how they contacted the first teachers and how the questionnaire was administered, it is therefore unclear what other limitations there might be in the methodology of the study. Theoretically, the group of teachers you interviewed could instead be representative of the total group, but it must be clear what the maximum number of teachers was.
The authors find below a table format; I recommend removing the vertical table borders before proof reading. In some numerical values, the comma (,) is left instead of the dot (.).
In the tables the authors must specify in brackets that the age is in years.
xx |
xx |
Xx (%) |
pvalue |
x |
x |
X (x) |
|
x |
x |
X (x) |
x |
x |
x |
X (x) |
|
x |
x |
X (x) |
|
x |
x |
X (x) |
|
x |
x |
X (x) |
|
x |
x |
X (x) |
|
x |
x |
X (x) |
x |
x |
x |
X (x) |
Do the authors comply to ICMJE recommendation for authorship?
In table 1:
-the title of the first column on the left is not specified
- "private" is a variable of the category "workplace" or "type of institution"?
-the numerical total of the categories is not 4622; if there are any missing values, they should be indicated, e.g. with an asterisk after the category "age" and then at the bottom of the table the authors should indicate how many values are missing
- what is the percentage referring to? it must be made clear what you are referring to (what is the total?)
in figure 1:
- It is visually clearer to leave spaces between the bars of different categories, as the authors have done in figure 2.
-there is a typo 'profl'
In figure 2 which variable are you referring to? It is not stated below the figure 'log[BF]'.
174-177 it sounds better in this other way 'in figure 1 it is described...'
182: is it [logBF<3.4] or [logBF>3.4]?
181-188 the paragraph is smoother to read if the authors write it like this 'age is strongly associated with [list of variables] and strongly associated with [list of variables]'. The same with age in lines 189-187. The strong association with age, what does it mean? In what way is there a difference between the young and the elderly?
257: what do the authors mean by points?
36-37: it is better to abbreviate “per cent” with “%” and “confidence interval x to y” with “CI x-y”
43: the authors meant “healthy workers”?
23: does the questionnaire have a name?
The introduction is a bit long; can the authors put little titles to the sections of the paragraph, so it is easier to frame the themes exposed?
The quality of the English is good; I noticed very few typos; some sentences are understandable but sound hard in English, for example: 136-137 sounds better in such way: “The aim of this study was to conduct an evaluation…” or “The aim of this study was to evaluate…”
I do not think there is a rule about the dot (.) at the end of the title, however, punctuation at the end of the title is generally only seen in the case of question marks or exclamation marks.

The quality of english is good but few sentences are a bit hard to read
Author Response
Thank you for your valuable comments. Please see the attachment.
1. Summary
Thank you very much for taking the time to review this manuscript. Please find the detailed
responses below and the corresponding revisions in track changes in the re-submitted files.
3. Point-by-point response to Comments and Suggestions for Authors
Comments 1: The manuscript entitled "Evaluation of the organisation of the COVID-19
vaccination process by the teachers in a region of Poland " investigates the opinions on the vaccination campaign of teachers in Poland. The population that was analysed is a very specific subgroup, yet the sample is relatively large. The topic is also still very timely. The results obtained are interesting and potentially applicable in other contexts, such as flu vaccination.
Response 1: Thank you for your valuable comments.
Comments 2: There are some issues that should be described in the methods section:
-Why did the researchers choose the teachers?
Response 2: Thank you for your valuable comments. The group of teachers was chosen for the study due to the fact that it represented the second professional group covered by vaccination in Poland after medical personnel. It therefore had a very high priority in the vaccination schedule in Poland, which was not the case until now. As mentioned in the introduction, such a high priority in the vaccination schedule for this professional group was not a common solution used by European countries. We have added this to the methods section to emphasize this point.
Comments 3: -teachers were found through the snowball effect, however at the beginning how did the researchers contact the first teachers?
Response 3: Thank you for your valuable comments. The researchers contacted the first teachers by directing email correspondence to the education departments of the provincial, district and municipal governments, then to the principals of the schools supervised by each local government, who finally forwarded it to the teachers. The first teachers then forwarded the link to the online form to more teachers. We have added this to the method section.
Comments 4: how many teachers did the researchers contact in total? So, what percentage of the teachers contacted answered the questions in the survey?
Response 4: Thank you for your valuable comments. As the authors did not contact teachers directly, but only forwarded correspondence regarding participation in the survey through their supervisors and superior institutions, it is difficult to estimate how many teachers were contacted by the researchers and what percentage of the teachers contacted responded to the survey questions. It can only be indicated that, according to statistics for Poland, 32489
teachers were employed in the Pomorskie Voivodeship in 2021. 4622 teachers participated in the survey, which represents 14.23% of this professional group. We have added this information in methods section.
Comments 5: did all the teachers give their consent?
Response 5: Thank you for your valuable comments. All surveyed teachers agreed to participate in the study. Consent to participate in the study was the first, mandatory element of the questionnaire. Failure to agree to participate in the study resulted in termination from the survey. We have added this information in methods section.
Comments 6: was participation compulsory? Was there a reward for participation?
Response 6: Thank you for your valuable comments. Participation in the study was voluntary and did not involve any rewards or compensation. We have added this information in methods section.
Comments 7: did the researchers follow any template or guidelines for writing this quantitative article?
Response 7: Thank you for your valuable comments. The researchers did not follow any template or guidelines for writing this quantitative article due to the lack of such studies for Poland in the subject area.
Comments 8: is there a particular reason why the researchers chose that region of Poland to conduct the research? Where is the region located and what is peculiar about it compared to others?
Response 8: Thank you for your valuable comments. The researchers chose this region of Poland to conduct this type of research due to the fact that they are professionally connected with this region of Poland on a daily basis, which had a significant impact on the possibility of easy access to the research group and obtaining research data. The surveyed region is located in the northern part of the country, by the sea, and is characterised by a good level of social and economic development compared to other Polish regions.
Comments 9: Why did the researchers choose that questionnaire?
Response 9: Thank you for your valuable comments. As mentioned earlier, for the purposes of this study, due to its specificity, the questionnaire was created from scratch and not selected from among the tools available in this area.
Comments 10: In the discussion section there is no interpretation of the meaning of the results, and consequently the comparison of your own results with those in literature.
Response 10: Thank you for your valuable comments. In the discussion, we added an attempt to interpret the obtained results (lines 317-325). However, due to the specificity of the study group and the previously mentioned fact that only a small number of European countries vaccinated this professional group, it is difficult to obtain comparative data.
Comments 11: The results are presented in the Results section, while in the discussion a commentary of the salient results should be provided without rewriting them again, both positive and negative, to try to identify possible meanings, patterns and relationships between your results and, if possible, between the results of other studies similar to yours.
Response 11: Thank you for your valuable comments. As mentioned above, in the discussion we added an attempt to interpret the obtained results (lines 317-325). However, due to the specificity of the study group and the previously mentioned fact that only a small number of European countries vaccinated this professional group, it is difficult to obtain comparative data.
Comments 12: For example: -how come in Figure 1 age and gender turned out to be so relevant?
Response 12: Thank you for your valuable comments. The lower assessment of the process of providing vaccination services observed among younger generations of teachers is probably related to the higher expectations of this group. However, the relationship between the obtained results and gender is probably related to the gender distribution in the study group, where in Polish conditions almost 80% of teachers are women. We have added this information in the discussion section.
Comments 13: in table 2 is there a directly proportional relationship between age of teachers and percentage of "very good" response for some items? How so?
Response 13: Thank you for your valuable comments. In the case of the “Level of patient service by vaccination center staff” assessment, there is a strong relationship between the assessment and the age of the respondents. In case of evaluation of “Information received at the vaccination center on possible vaccination reactions” and “Interview and qualification for vaccination at the vaccination center”, the relationship with age is very strong. Most respondents rated these elements very well, which may be due to the fact that vaccinations
for teachers took place in large, specialized hospital centers with experienced medical staff.
Comments 14: in table 3 many teachers answered: "hard to say." how come? Did they ever pay attention to the quality of information or transparency of rules?
Response 14: Thank you for your valuable comments. In this case, it is worth noting, first of all, that the assessments concern the general organization of the nationwide vaccination program, and not the vaccination process itself, which teachers had the opportunity to experience personally. This general organization of the vaccination program was most often known to teachers only from media reports, and perhaps due to the high dynamics of changes in the program, they may have suffered from a lack of knowledge or were unsure of their grades. However, they could express their opinions more radically regarding their own
experiences in the vaccination process, as presented in Table 2.
Comments 15: -what do the results obtained in figure 2 mean?
Response 15: Thank you for your valuable comments. In this analysis, we wanted to check whether the assessment of the vaccination process that teachers were directly subjected to was related to their assessment and perception of NPV as a whole. According to the data obtained, there was a strong relationship mainly between the assessment of individual NPV elements and the vaccination registration process. This is probably related to the fact that the process of
registering teachers for vaccinations was carried out via an IT system and was part of a nationwide whole. However, the remaining assessed elements of the vaccination process were related to elements shaped more regionally, i.e. in the vaccination facilities themselves. Interestingly, the strongest relationship in this analysis was between the assessment of information obtained at the vaccination point and the assessment of information within the central program relating to the management of post-vaccination reactions.
Comments 16: -do you think in table 4 some sources of information received quality unexpectedly? Such as online media having the same score as doctor, nurse? Or TV seems to give same opinions as government websites?
Response 16: Thank you for your valuable comments. Yes, in our Polish conditions, public television very strongly reflects and transmits exactly the same information that can be found on government websites. Therefore, their assessment may be very similar. In the case of similar assessments of the quality of information received from medical professionals and websites, it is worth bearing in mind that some teachers use very specialized websites, but
their collection was not the subject of analysis in this manuscript.
Comments 17: Are there any other questionnaires in the literature on this subject about
teachers? Knowing how to argue one's own results and those of other authors is important to put one's work in the context of the scientific literature (and thus the validity of one's work), whether the results agree or disagree.
Response 17: Thank you for your valuable comments. Unfortunately, as mentioned earlier, when designing this study, the authors did not find ready-made survey tools that could be used.
Comments 18: What are the implications of the study? In discussion the authors can reason about potential practical applications of the results.
Response 18: Thank you for your valuable comments. We tried to indicate the practical significance of the results of our research by changing fragments in the discussion (lines 358-368) and adding content in the conclusion (lines 427-436).
Comments 19: In the methods the authors vaguely mention where the questionnaire comes from. Can you include the website from which you obtained the questionnaire? Who designed it? In what areas was it tested? Why did the authors consider that questionnaire appropriate for your research question?
Response 19: Thank you for your valuable comments. As mentioned above, due to our
language error and lack of precision in the description, it may have seemed that we used a ready-made tool. However, we actually created the research tool from scratch.
Comments 20: In the introduction the authors describe the NVP; was vaccination mandatory or voluntary? Could the authors specify that in the methods section?
Response 20: Thank you for your valuable comments. Vaccinations under the NPV were voluntary. We have added this information in the method section.
Comments 21: In the introduction the authors need to stimulate the reader's interest and to highlight the importance of their work. Explain why research such as yours would be important in the past, current, or future context.
Response 21: Thank you for your valuable comments. The results of our research may be important from the point of view of future planning of vaccination programs in various professional groups. We have added this information in the introduction.
Comments 22: From your evaluation of your results and others in the literature, have you identified gaps in knowledge and that could be explored? The authors might, for example, suggest in discussion that future research should investigate teachers' opinions through qualitative research methodology.
Response 22: Thank you for your valuable comments. We have added this issue in the discussion section.
Comments 23: The results show that gender is associated with NVP evaluation. In what way? For example: in Figure 1, what difference would there be between males and females in the assessment? In table 3 are there any differences between genders?
Response 23: Thank you for your valuable comments. In the case of the data in Figure 1, gender showed a relationship with the NPV assessment. The detailed distribution of these results is presented in Table 3. In this table, gender differences occur only in a small part of the answers, where the BCI intervals do not overlap.
Comments 24: Normally in methods you do not describe how statistical tests work; in lines 154-161, the authors seem to be describing statistical concepts. I suggest the authors put the concepts expressed on log[BF] for example in this other way: "we considered values of log[BF] > 3.4 as suggesting a very strong relationship..".
Response 24: Thank you for your valuable comments. In this way, we wanted to give readers a little more insight into how our analyzes were carried out, which did not use traditional tools and tests. We have changed the indicated content in the methods section.
Comments 25: Among the limitations of the study, the authors report that the questionnaire is not representative of the group of teachers in the region due to the impossibility of conducting the questionnaire in situ during vaccinations. As a first piece of advice, do not say "is not a fully representative group", but instead it is better to say, "may not be fully representative". In this way you are reporting the limitation of your work without making it appear that you are demolishing the validity of your own sample.
Response 25: Thank you for your valuable comments. We have changed this sentence.
Comments 26: Since the authors did not report in the methods, however, how they contacted the first teachers and how the questionnaire was administered, it is therefore unclear what other limitations there might be in the methodology of the study. Theoretically, the group of teachers you interviewed could instead be representative of the total group, but it must be clear what the maximum number of teachers was.
Response 26: Thank you for your valuable comments. As mentioned above The researchers contacted the first teachers by directing email correspondence to the education departments of the provincial, district and municipal governments, then to the principals of the schools supervised by each local government, who finally forwarded it to the teachers. The first teachers then forwarded the link to the online form to more teachers. Moreover, as mentioned
above, according to statistics for Poland, the total number of teachers in the surveyed region was 32,489. Thanks to your comments, we have added this information to the methods section.
Comments 27: The authors find below a table format; I recommend removing the vertical table borders before proof reading. In some numerical values, the comma (,) is left instead of the dot (.).
Response 27: Thank you for your valuable comments. We have changed the formatting of the tables to the indicated ones. Additionally, we have unified the punctuation in the tables by using dots everywhere.
Comments 28: In the tables the authors must specify in brackets that the age is in years.
Response 28: Thank you for your valuable comments. We have added in the tables that age is expressed in years.
Comments 29: Do the authors comply to ICMJE recommendation for authorship?
Response 29: Thank you for your valuable comments. Yes, authors know and follow ICMJE recommendations regarding authorship.
Comments 30: In table 1: -the title of the first column on the left is not specified
Response 30: Thank you for your valuable comments. We have added the title for this column.
Comments 31: - "private" is a variable of the category "workplace" or "type of institution"?
Response 31: Thank you for your valuable comments. “Private" is a variable of the category "type of institution".
Comments 32: -the numerical total of the categories is not 4622; if there are any missing values, they should be indicated, e.g. with an asterisk after the category "age" and then at the bottom of the table the authors should indicate how many values are missing
Response 32: Thank you for your valuable comments. Missing values are indicated as suggested by adding an asterisk after the category name, and then at the bottom of the table it is indicated how many values are missing.
Comments 33: - what is the percentage referring to? it must be made clear what you are referring to (what is the total?)
Response 33: Thank you for your valuable comments. It is a percentage of all teachers surveyed (4,622). Added this information to the column name at the top of the table. Totals for each category have been added in the table.
Comments 34: in figure 1: - It is visually clearer to leave spaces between the bars of different categories, as the authors have done in figure 2. -there is a typo 'profl'
Response 34: Thank you for your valuable comments. Figure 1 contains much more data than Figure 2, so formatting it the same way is quite difficult. However, devoting an entire page solely to Figure 1 allowed for its improvement.
Comments 35: In figure 2 which variable are you referring to? It is not stated below the figure 'log[BF]'.
Response 35: Thank you for your valuable comments. As mentioned above In this analysis, we wanted to check whether the assessment of the vaccination process that teachers were directly subjected to was related to their assessment and perception of NPV as a whole. Due to the above, this analysis does not include the same variables as before, i.e., among others: age or gender.
Comments 36: 174-177 it sounds better in this other way 'in figure 1 it is described...'
Response 36: Thank you for your valuable comments. We have changed this sentence.
Comments 37: 182: is it [logBF<3.4] or [logBF>3.4]?
Response 37: Thank you for your valuable comments. It should be [logBF>3.4]. We have changed this to correct form.
Comments 38: 181-188 the paragraph is smoother to read if the authors write it like this 'age is strongly associated with [list of variables] and strongly associated with [list of variables]'.
The same with age in lines 189-187. The strong association with age, what does it mean? In what way is there a difference between the young and the elderly?
Response 38: Thank you for your valuable comments. The content of the indicated paragraphs has been changed as suggested.
Comments 39: 257: what do the authors mean by points?
Response 39: Thank you for your valuable comments. The authors had vaccination centers in mind. The sentence has been changed to be more precise.
Comments 40: 36-37: it is better to abbreviate “per cent” with “%” and “confidence interval x to y” with “CI x-y”
Response 40: Thank you for your valuable comments. We have changed this sentence.
Comments 41: 43: the authors meant “healthy workers”?
Response 41: Thank you for your valuable comments. Yes, it should be “healthy workers”. We have changed this sentence.
Comments 42: 23: does the questionnaire have a name?
Response 42: Thank you for your valuable comments. As mentioned above, this is a slip of the tongue and a lack of precision in the description of the tool. The study used an original questionnaire that does not have a name. We have changed this sentence.
Comments 43: The introduction is a bit long; can the authors put little titles to the sections of the paragraph, so it is easier to frame the themes exposed?
Response 43: Thank you for your valuable comments. We have added the titles to parts of introduction to more precise.
Comments 44: The quality of the English is good; I noticed very few typos; some sentences are understandable but sound hard in English, for example: 136-137 sounds better in such way: “The aim of this study was to conduct an evaluation…” or “The aim of this study was to evaluate…”
Response 44: Thank you for your valuable comments. We have changed the sentence as indicated.
Comments 45: I do not think there is a rule about the dot (.) at the end of the title, however, punctuation at the end of the title is generally only seen in the case of question marks or exclamation marks.
Response 45: Thank you for your valuable comments. We have removed the dot.
4. Response to Comments on the Quality of English Language
Point 1: The quality of english is good but few sentences are a bit hard to read
Response 1: Thank you for your valuable comments. We have subjected the content of our
manuscript to intensive linguistic editing.
5. Additional clarifications

Round 2
Reviewer 1 Report
I accept the paper based on the corrections.Needs work as noted in 1st review
Reviewer 2 Report
Thanks for the update, no more comments.